# Relying on the French territorial offer of thermal spa therapies to build a care pathway for long COVID-19 patients

Milhan Chaze[1], Laurent Mériade[1]*, Corinne Rochette[1], Mélina Bailly[2], Rea Bingula[3,4], Christelle Blavignac[5], Martine Duclos[4,6], Bertrand Evrard[3,4], Anne Cécile Fournier[7], Lena Pelissier[2], David Thivel[2], on behalf of CAUVIM-19 Group[¶]

1 University of Clermont Auvergne, "Santé et Territoires" Resarch Chair, CleRMa, Clermont-Ferrand, France, 2 University of Clermont Auvergne, CRNH, AME2P, Clermont-Ferrand, France, 3 CHU Clermont-Ferrand, Service d'Immunologie, CHU Gabriel-Montpied, Clermont-Ferrand, France, 4 University of Clermont Auvergne, INRA, UMR 1019, Clermont-Ferrand, France, 5 Centre Imagerie Cellulaire Santé, University of Clermont Auvergne, Clermont-Ferrand, France, 6 Service de Médecine du Sport et des Explorations Fonctionnelles, CHU de Clermont-Ferrand, Université Clermont Auvergne, INRA, UNH, Unité de Nutrition Humaine, CRNH Auvergne, Clermont-Ferrand, France, 7 Cluster Auvergne-Rhône-Alpes Innovation Innovatherm, Clermont-Ferrand, France

¶ Membership of the CAUVIM-19 research program is indicated in the "Acknowledgments" section.
* laurent.meriade@uca.fr

## Abstract

### Background

Work on long COVID-19 has mainly focused on clinical care in hospitals. Thermal spa therapies represent a therapeutic offer outside of health care institutions that are nationally or even internationally attractive. Unlike local care (hospital care, general medicine, para-medical care), their integration in the care pathways of long COVID-19 patients seems little studied. The aim of this article is to determine what place french thermal spa therapies can take in the care pathway of long COVID-19 patients.

### Methods

Based on the case of France, we carry out a geographic mapping analysis of the potential care pathways for long COVID-19 patients by cross-referencing, over the period 2020–2022, the available official data on COVID-19 contamination, hospitalisations in intensive care units and the national offer of spa treatments. This first analysis allows us, by using the method for evaluating the attractiveness of an area defined by David Huff, to evaluate the accessibility of each French department to thermal spas.

### Results

Using dynamic geographical mapping, this study describes two essential criteria for the integration of the thermal spa therapies offer in the care pathways of long COVID-19 patients (attractiveness of spa areas and accessibility to thermal spas) and three fundamental elements for the success of these pathways (continuity of the care pathways; clinical collaborations; adaptation of the financing modalities to each patient). Using a spatial attractiveness

**Funding:** This publication is related to the project named "CAUVIM-19 – Immuno-Metabolic (IM) Phenotyping and management of COVID-19: Specificity of actors and of the Auvergne territory". The "CAUVIM-19" project is co-funded by the FEDER (European Fund for Regional Development) as part of the European Union's response to the COVID-19 pandemic. The funders had no role in study design, data collection and analysis, decision to publish, or preparation of the manuscript.

**Competing interests:** The authors have declared that no competing interests exist.

method, we make this type of geographical analysis more dynamic by showing the extent to which a thermal spa is accessible to long COVID-19 patients.

## Conclusion

Based on the example of the French spa offer, this study makes it possible to place the care pathways of long COVID-19 patients in a wider area (at least national), rather than limiting them to clinical and local management in a hospital setting. The identification and operationalization of two geographical criteria for integrating a type of treatment such as a spa cure into a care pathway contributes to a finer conceptualization of the construction of healthcare pathways.

## Introduction

In November 2019, one of the largest pandemics of modern times began. Its global and rapid spread resulted a massive contamination of the world's population. According to official data from the American Johns-Hopkins University, the COVID-19 pandemic affected 559.5 million people worldwide and killed at least 6.4 million people by July 12, 2022. In France, the figures given by World Health Organization (WHO) reported 39,86 million cases of COVID-19 since the beginning of the pandemic (as of April 12, 2023) with a death toll of 162,176.

Far from the pandemic being over, contaminations are on the rise again. If the forms observed in the last few months seem less dangerous, the intensive research of the last two years has allowed us to highlight and document the numerous deleterious effects of the virus and in particular to focus on its long forms. For example, recent data have shown that a series of persistent symptoms can continue long after acute infection with SARS-CoV-2 has taken place. This condition is now referred to as long COVID-19 by scientists [1]. The National Institute for Health and Care Excellence (NICE) defines long COVID-19 as symptoms that persist or develop after acute COVID-19 infection and cannot be explained by any other diagnosis. In France, according to a study conducted by the national public health agency (*Santé Publique France)* last July 2022, the number of people who have been affected by a long COVID-19 is 2.06 million [2].

Knowledge of the many symptoms of long COVID-19 [3, 4] and the mechanisms that may explain them has improved significantly over the past two years. It must be noted, however, that the pathway of long COVID-19 patients after the acute phase has not been the subject of specific research on the type of care services likely to constitute a long COVID-19 patient pathway beyond strictly clinical treatment [5]. Indeed, the work on long COVID-19 patients has mainly focused on clinical treatments, ignoring the potential for innovation in the supply of care and the structuring of a pathway that this situation could allow to emerge at the territorial level [6]. The territorial level refers to a geographical area composed of health actors and structures with specific mobilizable resources. Because they can constitute major barriers to the follow-up of specific treatments, geography and space are essential determinants of the quality of care for long COVID-19 patients [6, 7].

Among specific treatments, thermal spa therapies represent a therapy offer located outside health care institutions (hospitals, clinics, medical centers) and whose integration in the care pathways of long COVID-19 patients seems to be little studied by researchers and considered by the health authorities. Important works have demonstrated, in different countries, that the medical management of COVID-19 patients favors the use of thermalism for its curative

effects related to post-disease pulmonary problems [8, 9] and for post-treatment rehabilitation [10–12]. Other studies remain more cautious about the benefits of spa treatments not combined with other treatments [13–15], particularly for the treatment of COVID-19 [16]. The medical benefits of spa treatments for long-standing COVID-19 also appear to be the subject of intense academic debate. However, irrespective of the outcome of this medical controversy, few studies have focused in parallel on an important practical and organizational issue: the potential access of patients to these spa treatments. Indeed, in contrast, we do not find any work analyzing the ways in which it is possible to integrate these thermal therapies into the care pathway of long COVID-19 patients. Indeed, beyond the health effects of thermal spa therapy on COVID-19 patients, it seems essential, in parallel, to put in perspective the possibilities of thermal spa therapy offered by the territories and their integration in an adapted care pathway. This is why our research focuses on patients' access to this resource (thermal treatments) and the possibility of integrating them into a post-treatment pathway of the acute phase of the disease if the results of ongoing studies (such as CauvimTherm in France but not yet published) show that there are beneficial conclusive effects.

## What place can thermal spa therapies take in the care pathway of long COVID-19 patients?

This is the question that has guided our reflection and has led us, in the case of France, to explore the place that thermal spa therapy could take in the care pathway of long COVID-19 patients. To carry out this reflection, we mobilized geographic mapping in health care. In health care pathway management studies, geographic mapping is still under-exploited [17], probably because of the limitations associated with it: idealized [18] and reductive [19] representations, the ideological imprint of its sponsors and/or designers [20], and the impossibility of representing on the scale of a geographical space in all its completeness [21]. So far, attempts to geographically conceptualize care pathways [22–24] have used the map as a descriptive and static tool at the service of public health policies and less at the service of patients and health professionals [13, 25]. Indeed, geographical mapping is mainly used to visualize national and sub-national health data [26]. In public health, geographical mapping is sometimes used collaboratively [27–29] or thematically [30], but without really superimposing healthcare needs and resources integrating a care pathway. On the one hand, cartographic interaction and collaboration, representing the dialogue between a human and a map, provides auxiliary background knowledge and auxiliary map-reading tools to facilitate the transfer of geographical knowledge to the public [29]. On the other hand, these approaches to geographical analysis remain less dynamic and prescriptive in terms of care pathway management.

We propose here to use geographical cartography to carry out an interactive and cross reading of the possible care trajectories of long COVID-19 patients and of accessibility to the French territorial offer of thermal spa therapies. The objective of this reading is, from the geographical cartography, to determine what place French thermal spa therapies can take in the care pathway of long COVID-19 patients. Indeed, unlike proximity care (hospital care, general medicine, para-medical care), the thermal cures present national or even international accessibility. The models of spatial accessibility to the care based on the proximity between patients and care only partially answer the question of the integration of the thermal treatments in a care pathway. Also, in order to answer our research objective, we build a geographical cartographic analysis in two stages by mobilizing in particular the method of evaluation of the attractiveness of an area defined by David Huff [31] which allows measures of accessibility to care to be widened to a national or even international patient base.

To report on this study, our article is structured as follows. Firstly, we review the main studies on long COVID-19 and its consequences on the care pathways of affected patients. Secondly, we present our geographical mapping methodology. Thirdly, we present the main maps illustrating the application of geographical mapping analysis integrating Huff's attractiveness model. Finally, we discuss these results and highlight their main contributions to the management of care pathways in general, and more specifically to the management of long COVID-19 patient pathways.

## Background

Knowledge about the persistent symptoms and rehabilitation needs of long COVID-19 patients has begun to emerge [32–34]. The most common persistent symptom was fatigue (53% to 64%), followed by dyspnea (42% to 50%) [34–36]. Other symptoms were psychological distress, joint pain, chest pain, cough, sleep disturbance, and functional disability. Patients also reported a decrease in quality of life [32–34]. The management of these symptoms requires mobilizing a wide range of resources to manage the sequelae of long COVID-19 patients [37]. The National Institute for Health and Care Excellence (NICE) guidance in the United Kingdom highlights the value of multidisciplinary rehabilitation in managing the post-COVID-19 patient pathway [37]. There is an interest for more relevance in setting up individualized and adapted rehabilitation programs to meet patients' needs [38].

Therapeutic effects of spa treatments on chronic pathologies through the reduction of pain, the improvement of patients' comfort and the reduction of medication have been demonstrated [39]. Twelve therapeutic orientations of spa medicine are recognized in France by the health insurance system and entitle patients to reimbursement for a treatment. These orientations and the resulting treatments help to alleviate the effects of a chronic pathology. They concern, among other pathologies, rheumatological affections, respiratory tracts, psychosomatic affections, which are also pathologies identified within the framework of the symptoms reported by long COVID-19 patients.

Research into the manifestations of infection and their effects points to symptoms that are in part, symptoms that spa treatments can act upon. Indeed, recent evidence shows that various treatments using thermal water are effective for several diseases of the respiratory tract [9]. Consequently, the spa environment could represent an appropriate out-of-hospital setting for respiratory rehabilitation in post-COVID-19 subjects. However, further studies are needed to test the efficacy of spa respiratory rehabilitation protocols for these patients [9] to support the initial results of the studies [39].

For example, respiratory treatments (based on inhalations or gargles, prescribed and supervised by specialized health personnel) stimulate the immune system and clean the respiratory system, preparing it to face a possible episode of COVID-19 [35]. Several studies [9–11, 40] underline the interest of offering rehabilitation interventions based on the thermal spa therapy offer for COVID-19 patients suffering from musculoskeletal and neurological disabilities characteristic of long COVID-19 patients. Recently, the study conducted by Gvozdjáková et al. [41] would tend to demonstrate that a high-altitude environment accompanied by spa rehabilitation can be recommended to accelerate the recovery of patients with long COVID-19 syndrome.

A thermal spa therapy is an ideal space to recharge one's batteries between primary care and Physical Medicine and Rehabilitation centers [42]. It allows the patient to leave their normal situation and habits, offers essential rest time during the convalescence phase, allows the re-appropriation of chronobiological rhythms including sleep which is the pillar of recovery, the resumption of an adapted physical activity and resocialization [43]. Other studies also

show that balneotherapy has thermal effects on the body and may present certain dangers for patients [13, 14, 16]. Other studies show a potential effect of spa treatment in the medium term rather than the short term, but always in combination with other therapies [15, 35, 40, 41]. Consequently, there is no total consensus on the benefit/risk ratio of spa treatments for long COVID-19 patients.

Thermal spa therapy constitutes a unit of time and place conducive to learning self-management of the disease [43]. It is characterized by a holistic and person-centered approach, led by a multi-professional team (doctor, physiotherapist, nurse, dietician, psychologist, hydro therapists) experienced in the practice of thermal spa therapies. Thermal spas can also sometimes offer the advantage of being inexpensive therapies for health insurance compared to medical drug management.

Antonelli & Donelli [12] propose a standard/core care model of COVID-19 patient management (Mechanical Pulmonary Ventilation for Rehabilitation, Mineral Water Inhalation Therapies, Physical Activity, Psychological Support) that refers to already existing rehabilitation plans with a long tradition, such as those prescribed for work-related respiratory diseases such as pneumoconiosis, whose long-term outcomes share some clinical features with post-infectious pulmonary fibrosis.

The need to start respiratory therapy as soon as possible, while hospital resources are over mobilized by acute management [44], and the facilities available to treat patients in the post-acute period being limited [17] local thermal spa facilities constitute an appropriate setting for respiratory rehabilitation interventions in post-COVID-19 subjects [45]. If thermal spa therapies can bring benefits on the physiological level (respiratory discomfort, musculoskeletal pains) they also present an interest at the psychological level (anxiety, sleep disorders) by acting on well-being and can be part of a global approach to the treatment of long COVID-19 symptoms. Thus, for example, a broad review of the literature on the therapeutic effects of thermal spa therapies allows Castelli et al [46] to hypothesize the mutual and reciprocal effects of these therapies on pain reduction and sleep quality.

It is in this perspective of global treatment that thermal spas have developed a long COVID-19 offer but the care pathways in a patient-centered territorial proximity offer remain to be thought out [17]. A geographical mapping of the thermal spa offer and an analysis of its attractiveness/accessibility to potential patients can help to develop care pathways for post-acute COVID-19 patients. Indeed, maps are regularly used in public health as secondary methodologies for representing primary location data cartographically [47]. However, they are rarely used to dynamically read aggregated geographical data in order to manage care pathways.

## Materials and methods

Our research is non-interventional and therefore not subject to the rules of the Jardé Law (Law no. 2012–300 of March 5, 2012) on biomedical research carried out in France, since it does not use any personal data relating to the health of a specific cohort of patients. Regarding personal data protection, in the European Union and in France, the General Data Protection Regulation (GDPR) (Regulation 2016/679 of the European Parliament and of the Council of April 27, 2016) came into force on May 25, 2018. The French Data Protection Act (Law no. 78–17 of January 6, 1978 on data processing, data files and individual liberties) came into force on May 25, 2018.

These two texts now form the basis of the new regulations on personal data protection. The GDPR defines personal data as "any information relating to an identified or identifiable natural person", i.e. a natural person who can be identified, directly or indirectly. In practice, this

can mean identifying data such as surname, first name, address or telephone number, information relating to the patient's personal life (e.g. number of children), social security coverage (e.g. compulsory health insurance, supplementary health insurance, etc.) and, above all, information relating to the patient's health (pathology, diagnosis, prescriptions, care, etc.), as well as to the professionals involved in his or her care.

In our study, we do not use any of these data, as they are only statistical data from open national databases (INSEE, Santé Publique France, Conseil National des Etablissements Thermaux) concerning, on the one hand, the proportion of patients affected by a long COVID-19 and their geographical locations and, on the other hand, the socio-economic characteristics of French thermal spas as well as their geographical locations (see S1 Appendix).

Seven hundred and seventy natural mineral water springs are listed in France, i.e. 20% of the European thermal capital, which gives it first place in terms of European hydromineral heritage. Ninety thermal areas are in operation in France (see S2 Appendix)., for 110 thermal spas (cf. French thermal spas, National Council of Thermal Establishments, Edition 2022). Three major regions (Occitanie, Auvergne-Rhône-Alpes, New Aquitaine) account for 80% of French thermal spa therapy centers (Curist survey 2006, National Council of Thermal Establishments).

In order to put into perspective, the opportunities offered by the French thermal spa therapy offer for the management of long COVID-19 patients, we proceeded to a geographic mapping analysis of the available official data on COVID-19 contaminations, hospital admissions of COVID-19 patients in intensive care and available data on the national offer of thermal spa therapies by relying on an analysis of the offer of all thermal spas. This analysis was carried out in two successive and cumulative steps.

## Data for the geographic mapping analysis

In order to carry out our geographical cartography, we used the geographical cartography software Chronomap®.

Firstly, to build this geographic mapping, statistics on the French territorial organization and population of each region are taken from official data from the National Institute of Statistics and Economic Studies (INSEE) [48]. Statistics on COVID-19 positive cases and COVID-19 cases requiring admission to an intensive care unit are based on official French data published by *Santé Publique France* (national public health agency) [49] They cover all COVID-19 cases and admissions to intensive care units between March 19, 2020 (start of the census of cases in France) and May 30, 2022.

Regarding attendance at thermal spas, the data we have processed are those of the National Council of Thermal Establishments (CNETh) [50] which includes all French thermal spas whether they are public, private, for profit or not for profit. The choice of the year 2019 to measure the number of thermal spa patients is justified by its anteriority to the COVID-19 epidemic which strongly disrupted thermal spa therapies during the year 2020. We also used the data of specialization of thermal spas (*cf*. location-cure.net, guide to French thermal spas from the National Council of Thermal Establishments) as well as the geocoding in two points (longitude/latitude) of the whole of the thermal spas counted.

## Analysis method of attractiveness and accessibility to thermal care

Secondly, we applied an analysis of the attractiveness and spatial accessibility to thermal spas for the potential patient concerned by this type of care. We chose to apply our method on spas whose specialties are likely to enter the care protocol of long COVID-19, in order to get a global view point on the offer. However, we distinguished the medical specialities in order to

show their location in France. Various measures of spatial accessibility are proposed, including regional availability [47], the gravity model [51, 52] and the 2 Steps Floating Catchment Area (2SFCA) method [53].

The regional availability method performs a simple relationship between supply (physicians) and demand (population) within a predefined area (usually the "Department"—in France the "Department" is one of the three levels of government under the national level, between the administrative regions and the communes. 96 departments are in metropolitan France, and 5 are overseas departments). However, this method does not reveal spatial variation within the boundary and does not account for the interaction between supply and demand across the boundary [53]. The gravity model is theoretically more robust, but it requires more computation and the result is not intuitive to interpret [54]. In this model, the interaction data between the place of care and the patient are often specific to a region [55], which is relatively unsuitable for a thermal spa whose area of attraction goes far beyond the borders of a region. Among the gravity methods, the 2SFCA method retains most of the advantages of a gravity model [56] and generates a physician/population ratio that is determined primarily at the local catchment or regional level [51]. Despite its limitations, the 2SFCA method is the most widely used to measure spatial accessibility to health care [49]. However, it is a dichotomous measure where all locations outside the catchment area are assumed to have no access [56]. This mode of calculation is not well adapted to the thermal spas which, unlike proximity care (local doctors, pharmacies, etc.) have a national or even international attractiveness and not only local.

The method of evaluation of the attractiveness of an area defined by Huff (also known as Modified Huff Model three-step floating catchment area—MH3SFCA) [31, 56, 57] is also a gravity model but it allows us to consider the distance between the supply and the demand whatever the zone of influence of the care place. It is a spatial interaction model based on the gravitational principle, i.e. that the attractiveness of an area is proportional to the offer we are talking about (the capacity of the thermal spas in this study) and inversely proportional to the distance which separates it from the patient base. The application of this model allows to define the degree of attractiveness of a place by another without being limited to a local health basin. This is why it is well suited to the particularities of our study. In the MH3SFCA method, a population's demand only decreases with increasing distance from a service site if other service sites are available. The MH3SFCA method is therefore highly sophisticated, combining many of the advantages of previous traditional methods (regional availability and gravity model) with the benefits of more advanced methods (2SFCA) [58].

We have applied this method in two steps of calculation. In a first step, we calculate the probability that the patients of each department carry out their cure in the various thermal spas according to the capacity of reception of each thermal spa and of the distance of it to the department, by applying the following formula:

$$P_{ij} = \frac{W_i / D_{ij}^a}{\sum_{i=1}^{n} \left( W_i / D_{ij}^a \right)}$$

With:

Pij: Probability that patients from department j will take their treatment in thermal spa i.
Wi: Capacity of the thermal spa i.
Dij: Distance between the centroid of department i and thermal spa j.
a: Exponent applied to reduce the probability of distant sites.

The capacity of the thermal spas was determined by the number of patients in 2019, in order to avoid the year 2020, marked by the epidemic of COVID-19 which could have

disrupted the thermal frequentation. The choice of spa attendance in a typical year rather than the capacity of the thermal spa allows us to overcome the bias of the variable duration of the cures, which also influences the actual attendance of the spas.

The distance between the patients and the thermal spas was calculated from the address of each spa and the centroid of the departments. The choice of the department centroid is explained by the departmental scale at which the data on patients treated for COVID-19 were obtained. The distance between the two points was calculated as the crow flies, and not by road, in order to take into account, the insular nature of Corsica, where access to the island is by boat, but also by plane. The choice of the bird's eye view, even if it is less precise than road accessibility, allows us to measure accessibility in the same way for each area. The application of this formula has thus allowed, in a first instance, to calculate the attractiveness of each department to each thermal spa.

In a second step, in order to determine the attractiveness of each area for all thermal spas, we have added the previously calculated indices, obtaining an accessibility indicator of each area for all thermal spas. We have carried out the same operation for the thermal spas in order to obtain an attractiveness indicator considering not only their capacity of reception, but also their geographical situation. The formula of the second step is the following one.

For the accessibility of each department:

$$A_j = \sum_{i=1}^{n} P_{ij}$$

For the attractiveness of each thermal spa:

$$A_i = \sum_{i=1}^{n} P_{ij}$$

With:

Aj: Accessibility of the department j to all the thermal spas.

Aj: Attractiveness capacity of the thermal spa i for the whole departments.

The result of this analysis of accessibility has finally been represented on a map with double reading: by the attractiveness of spas areas, and by the accessibility to the thermal spas.

To produce the maps presented in this article, we used a background map of the French Departments from Global Administrative Areas Data and Maps (GADM: https://gadm.org/). These backgrounds are royalty-free and compatible with the CC-BY-4.0 license used by PLOS, as indicated on the GADM website (https://gadm.org/license.html). We then imported this base map into the GIS software QGIS (CC-BY-3.0 license: https://www.qgis.org/fr/site/). We then attached our statistical data to the base map to map the *INSEE* and *Santé Publique France* data. Spa centers were geolocated using their addresses, enabling us to create the second layer of our GIS. We then joined the spa data to the points representing them on the map and were thus able to map the spa data. Finally, we calculated each departments' potential access to spa centers, as described in the article's methodology.

## Results

### The thermal spa therapy offers likely to intervene in the care pathway of long COVID-19 patients

The French thermal spa therapy offer presents, for natural and historical reasons, a very unequal distribution. Four main areas appear: the Pyrenees and their foothills, the Northern Alps, the Massif Central and the Vosges. In addition to these areas, there are scattered centers (Balaruc in the Hérault province, Amnéville in Moselle province, Niederbronn in Alsace province, Saint-Amand-les-Eaux in the North, Bagnoles-de-l'Orne. . .). The Pyrenees and their

foothills are the main focus, but the Auvergne-Rhône-Alpes region, with the Auvergne and North Alpine thermal spas, have the second largest spa therapy offer in France (Fig 1).

The activity of thermal spas is also very uneven. Through the analysis of the number of spa patients received by each of them in 2019 (the year preceding the COVID-19 epidemic), some large centers emerge: Balaruc, Aix-les-Bains and Dax clearly stand out. Dax and its suburbs (Saubusse-les-Bains, Préchacq-les-Bains, Tercy-les-Bains) also has the originality of having 22 thermal spas, 16 of which are located in Dax alone (a record in France). That being said, the majority of thermal spas are small entities, more than half of which do not welcome more than 5,000 patients per year.

In terms of specialization, most French spas are specialized in rheumatology and a good half of them combine this specialty with ENT (Ear-Nose-Throat) and respiratory problems (see S1 Appendix). In contrast, very few French spas are specialized in psychosomatic pathologies (S1 Appendix), but, as previous studies have revealed [38, 43], the treatments of these pathologies undoubtedly benefit from rheumatological or respiratory treatments.

On the one hand, thermal spas with the highest capacity (Dax in the South-West of France, Balaruc-les-Bains in the South or Aix-les-Bains in the Alps) are rather specialized in rheumatology (Fig 1). On the other hand, the thermal spas specialized in ENT and respiratory affections, very concerned by the treatment of the symptoms of the long COVID-19, are not big thermal spas in terms of capacity of reception (Fig 1). Moreover, most of the thermal spas specialized in these pathologies are located in the Pyrenees and their foothills and in the region Auvergne-Rhône-Alpes (Alps and Massif Central—La Bourboule, Le Mont-Dore, Allevard, Saint-Martin-d'Uriage, Saint-Gervais-les-Bains. . .). Only a few spas are more isolated (Amnéville, Saint-Amand-les-Eaux, Jonzac).

## Thermal spa therapy and COVID-19 contamination sites

If we compare the thermal spa therapy offer with the geographical distribution of COVID-19 cases, some major points appear. First of all, several important foci of COVID-19 contamination appear (Fig 1), in particular Ile-de-France, Auvergne-Rhône-Alpes, Provence-Alpes-Côte d'Azur, Toulouse Region, Nord-Pas-de-Calais. The location of the main sources of contamination corresponds to the main urban concentrations (Paris Region, Lyon Region, Marseille, Toulouse, Nord-Pas-de-Calais), as well as to the initial sources of contamination (Val-d'Oise in the Paris Region, Rhône Region, Northern Alps, Alsace).

However, this geography of infections must be complemented by the geography of severe cases that required admission to intensive care, which includes the majority of long COVID-19 cases (Fig 2).

Admissions to intensive care are mainly in regions with large urban areas (Paris, Lyon, Marseille), both because of the concentration of population and the presence of major hospital infrastructures. In addition, there is a clear contrast between a north-east-south arc where contamination and admissions to intensive care are significant, and a more spared west-central area.

The interactive reading of the geography of the French thermal spa therapy offer and of the COVID-19 contaminations/severe cases (Figs 1 and 2) allows us to enrich very appreciably our way of analyzing and managing the care pathway of long COVID-19 patients. Indeed, on the one hand, this cartographic analysis indicates that the thermal spas located in the heart of, or near the main areas of contamination, and in particular the resuscitation admission centers, appear to be adapted structures in terms of accessibility, for patients suffering from long COVID-19 and are therefore favored. Indeed, the Alpine thermal spas (Aix-les-Bains, Allevard, Saint-Gervais-les-Bains. . .), located in the heart of the Rhone-Alpine contamination

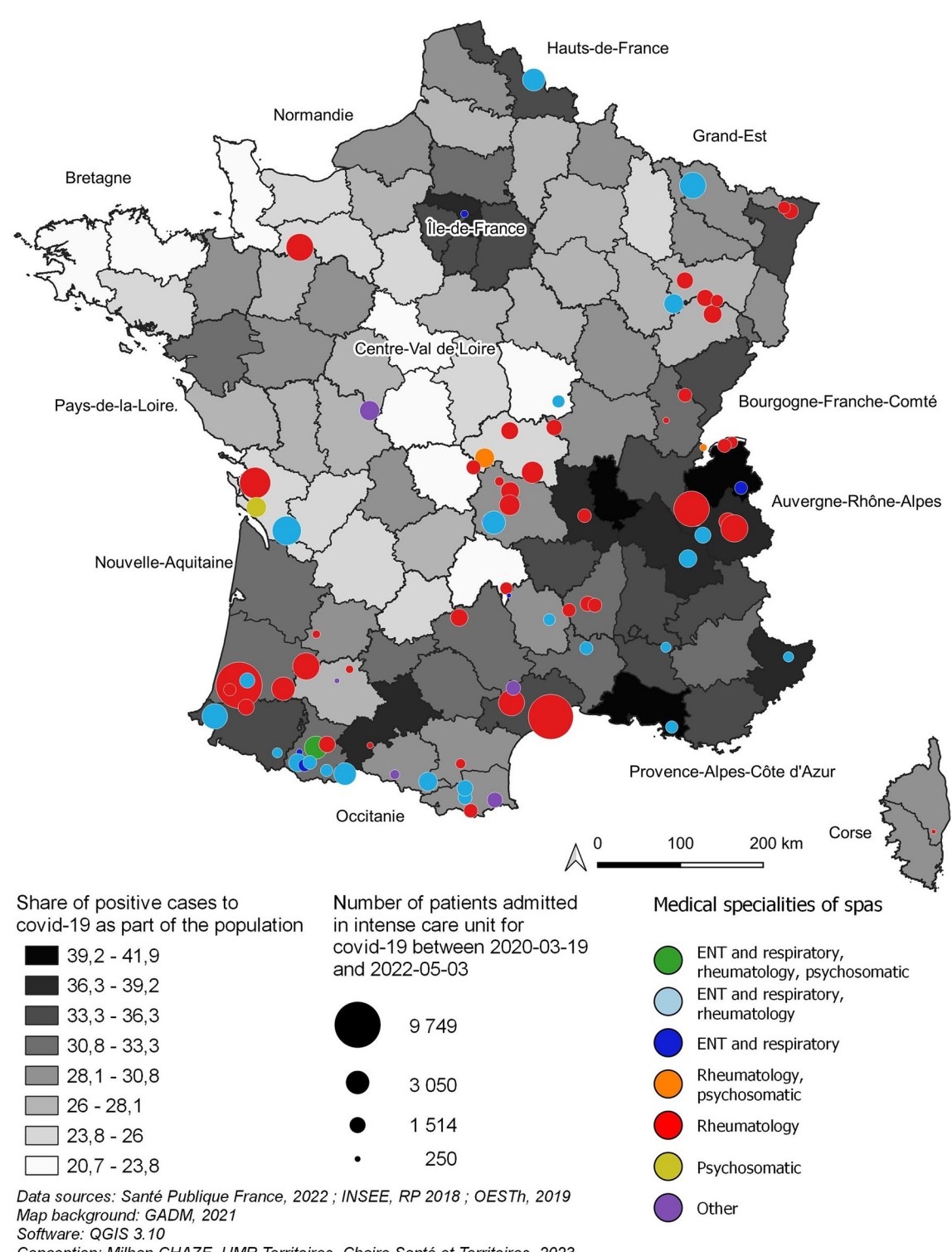

**Fig 1. Attendance and specialisation of thermal spas (2019) and positive cases to COVID-19 (2020–2022).**

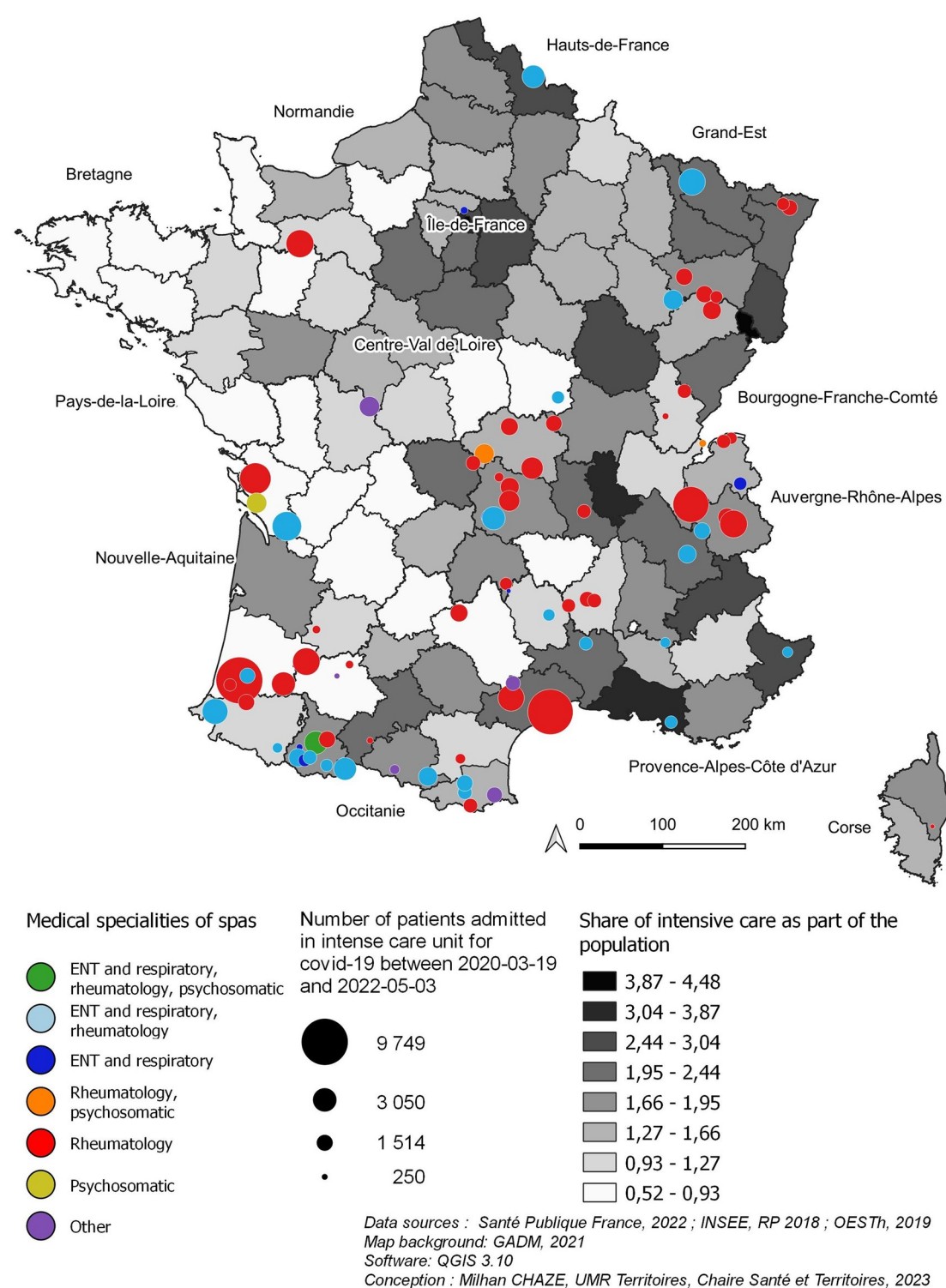

**Fig 2. Attendance and specialisation of thermal spas (2019) and rate of admissions in intensive care for COVID-19 (2020–2022).**

area, are particularly well placed to receive higher numbers of patients. This is also the case, to a lesser extent, of the Pyrenean thermal spas in relation to the Toulouse region.

On the other hand, some regions, such as Ile-de-France and Nord-Pas-de-Calais, with few thermal spas but strongly affected by COVID-19, constitute important areas of potential patients for thermal spas located relatively far away. The economic support of the long COVID-19 patients of these distant regions, the conception of thermal stays adapted to their needs and more important collaborations between the public administrations of these regions and the thermal spas already appear as levers of reduction of these inequalities of access very adapted to the French socio-economic specificities.

However, this cartographic analysis also identifies the need, as far as thermal cures in particular are concerned, to measure the level of accessibility of potential patients to these cures which constitute care whose attractiveness largely exceeds the borders of a medical area of proximity or of a region.

## Attractiveness and accessibility to the thermal spas

The departments with the highest accessibility to the thermal spas are those which are the closest to the thermal spas. However, thanks to the method used, the proximity can be relativized. Some departments like Gironde (Nouvelle Aquitaine region) or Lozère (Occitanie region) or Meurthe-et-Moselle (Grand Est region), which don't have a thermal spa on their territory, or only a small center like Lozère, benefit from a very good accessibility to the thermal offer because of the presence of a highway which connects them correctly to the stations (Fig 3).

In fact, we find areas of high accessibility in the South-West of France, in the North-East, as well as in the heart of the Massif Central and in the Northern Alps. On the other hand, the heart of the Paris Basin, Brittany, Provence-Alpes-Côte d'Azur and Corsica present the weakest potential of access to thermal spas. The main explanation lies in the unequal geographical distribution of the thermal spas. But to this, we have to add the unequal capacity of attraction of thermal spas which reinforces the potential of access of some departments like Hérault (Occitanie region) thanks to the presence of Balaruc (the most important French thermal spa), Savoie (Auvergne-Rhône-Alpes) with Aix-les-Bains, and of course the departments of the South-West around Dax and the more modest thermal spas of the Pyrenees.

In addition, the geographical location of a thermal spa within a department also influences its accessibility. This is the case for Saint-Amand-les-Eaux in the Hauts-de-France, whose relatively central geographical position allows it to be easily accessible from the entire department (Fig 3).

If we relate the accessibility of the territories to the thermal offer with the geographical distribution of the patients having had long COVID-19 (Figs 1 and 2), a real disparity between the foci of COVID-19 and long COVID appears as for their accessibility to the thermal spas. Indeed, Ile-de-France, and to a lesser extent the South-East of France, which constitute important foci of COVID-19, are relatively far from the thermal spas, whereas other important foci such as the Lyon region and the Northern Alps (Auvergne-Rhône-Alpes) are well served by the thermal spas.

These inequalities of accessibility risk adding an additional cost to the expenses of the patients for their travel to the thermal spas. In fact, some thermal spas located near theirs home locations could be preferred by the patients living nearby, and this, in spite of a sometimes limited capacity of reception. This could be the case of the thermal spas located at the limits of the Paris Basin, namely the most accessible ones of the Paris Region where we find an important concentration of patients treated for a COVID-19.

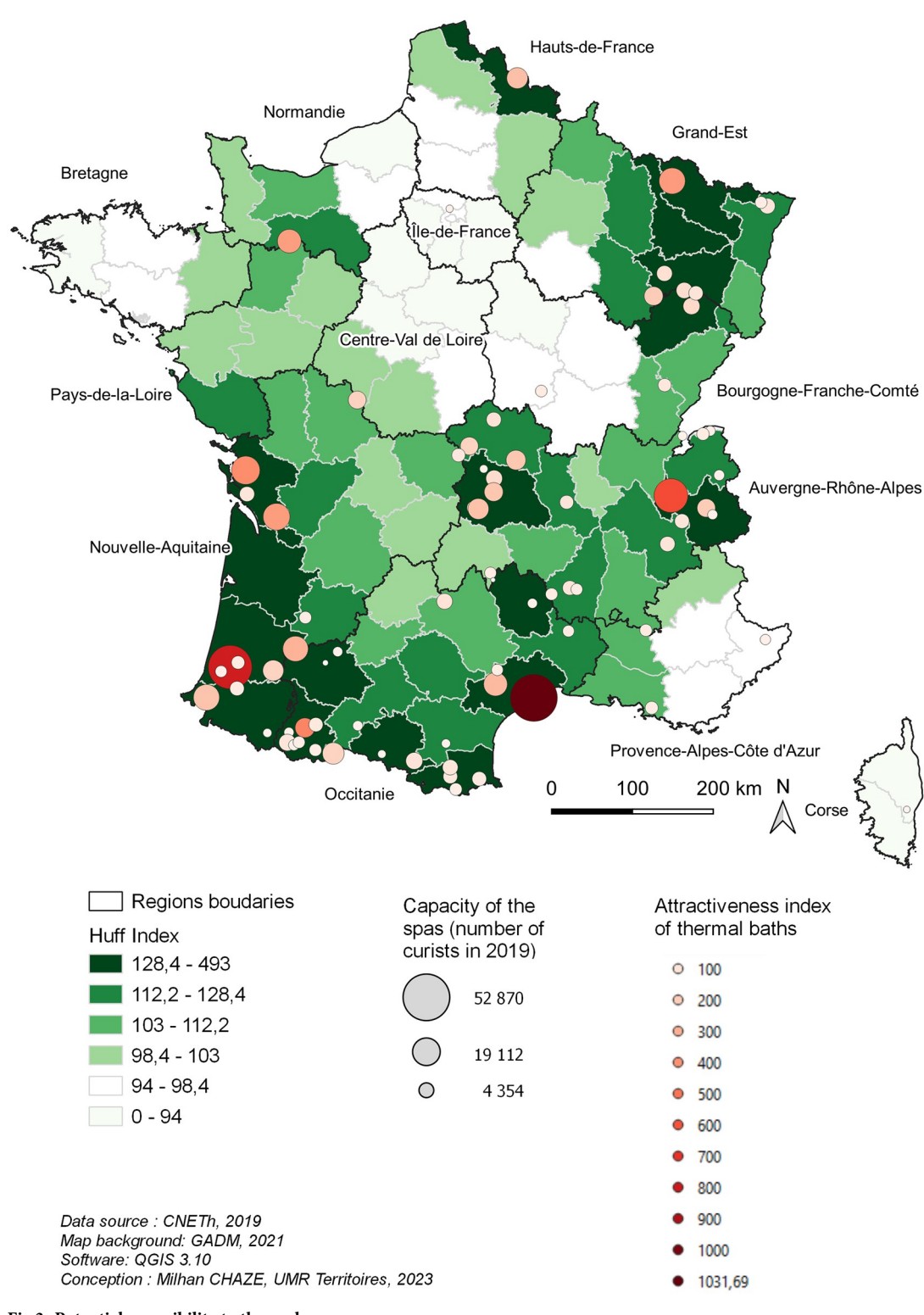

**Fig 3. Potential accessibility to thermal care.**

The case of Ile-de-France (Paris) contrasts with that of the other large French conurbations (Lyon, Marseille, Toulouse, Bordeaux), which constitute other major foci of COVID-19, but whose accessibility to the thermal spas appears to be better even though they do not have thermal spas immediately nearby (Fig 3).

Huff's model [31, 56], which combines the location of potential patients with the attractiveness of spa areas, significantly strengthens the analysis of accessibility to thermal spas, thus promoting the integration of thermal treatments into the care pathway of long COVID-19 patients.

## Discussion

### Analysis of thermal spas accessibility reinforced by the Huff Model

Several ideas emerge from the cartographic analysis of the accessibility to the thermal spas realized from the method of Huff [31, 56]. Our results show that, when it comes to integrating spa therapies into patient care pathway, the application of Huff's method offers important theoretical and managerial perspectives. Indeed, as mentioned above, the regional availability method [47], the gravity model [51, 52], and the 2 Steps Floating Catchment Area (2SFCA) method [53] mainly allow the location of a local patient to be considered. For spa centers, however, the patient base is national or even international. Our results therefore show that Huff's method offers the possibility of taking this national patient base into account and therefore facilitates the integration of spa therapies into the care pathway of patients with long COVID-19.

Thanks to Huff's method, it is possible to distinguish thermal spas according to their respective levels of attractiveness and accessibility. In the first place, the spa areas which present the highest potential of attraction are the ones which have the most important capacity of reception, and this, independently of the distance the geographical situation of the spas, because of a scale effect. As the study concerns all the thermal spas of a same department, the longest distances are always compensated by the shortest ones for each thermal spa.

Indeed, whereas the cartographic analysis would tend to show a relative distance of the COVID-19 foci from the thermal spas (Figs 1 and 2), the Huff's model relativizes appreciably this distance by demonstrating that the accessibility to the thermal spas is good for a good part of the south and the east of France. Only the center of Northern France (Regions, Ile-de-France, Hauts-de-France, and Centre) presents a distance that we can qualify as important. This analysis therefore sheds light on the question of patient accessibility to care that is attractive nationally and even internationally. It allows a much finer reading of the accessibility to care (here thermal) which considers at the same time, the care offer of a territory and the location of the patients presenting a single and same pathology. Therefore, this cartographic and accessibility analysis makes it possible to adapt the care pathway of the COVID-19 patients according to the potential accessibility of the patients to the thermal care and not only according to their geographical proximity to the thermal spas.

### Adapting care pathways to the location of patients and the thermal spa therapy offer

In France, thermal spa therapies prescribed by a physician are reimbursed at 65% by the health insurance (the rest is covered by the patient's private health insurance) and at 100% for patients suffering from an occupational disease (for a maximum duration of 18 days) (cf. French thermal spas, National Council of Thermal Establishments, Edition 2022). The thermal spa is chosen by physicians with the patient's agreement (or on the patient's proposal in 50% of the case) according to the patient's pathology and is not necessarily the one closest to the

insured person's home for the prescribed orientation. Accommodation and travel costs are the responsibility of the patient. They can also be covered only if they have insufficient resources. In fact, transportation and accommodation are only covered for patients whose annual income does not exceed 14,664 euros per year for a single insured person. For a married person without children, this ceiling rises to 21,996 euros [59]. However, if the insured person is eligible for the supplementary transport benefit, the reimbursement will be based on the distance separating the patient's home from the nearest thermal spa suitable for the therapy. Also, in practice, the conditions of coverage of the spa therapy by the health insurance and especially the necessity for some patients to finance their accommodation and their transport lead doctors to choose the thermal spa closest to the patient's home. From then on, as the geographical cartography suggests, the proximity between the territorial offer of thermal spa therapies and long COVID-19 patients constitutes an essential determinant of the integration of thermal spa therapies in the care pathway of these patients. The South of France (Auvergne-Rhône-Alpes Region, Provence-Alpes-Côte d'Azur Region, Occitanie Region, New Aquitaine Region) represents both 80% of the French spa therapy offer and the French regions with the highest COVID-19 contaminations between 2020 and 2022 (Fig 1). Among the thirteen French regions, these four regions represent geographical areas in which thermal spa therapy can be integrated with ease into the care pathway of COVID-19 patients. This result is particularly striking in the two regions of the South-East of France (Auvergne-Rhône-Alpes Region, Provence-Alpes-Côte d'Azur Region) where the importance of severe cases of COVID-19 which geographical location is very similar with the thermal spa therapy offer (Fig 2). In these last two regions, the thermal spa therapy can be integrated in the care pathway of the long COVID-19 patients without any important geographical obstacles.

## Examining the collaboration between healthcare stakeholders for patient's health

In these regions, the proximity between the territorial offer of thermal spa therapies and the location of the patients authorizes a closer collaboration between hospital physicians, general practitioners and thermal spas. The development of this collaboration is an even more interesting prospect as thermal spas can, in parallel, increase the number of rehabilitation places for long COVID-19 patients and, consequently, reduce the workload of hospital rehabilitation units [60].

On the other hand, in the other French regions where the thermal spa therapy offer is reduced but in which COVID-19 contaminations or the number of serious cases are high (Paris Region and North-East of France) (Figs 1 and 2), the integration of thermal spa therapy in the patients' care pathway seems more constrained at the geographical level. This integration is due to very unequal care pathways on the socio-economic level between high income patients able to finance all or part of their treatment, accommodation and transport and those only able to finance a part of this treatment. Also, it is undoubtedly necessary to think of a mode of financing and the reimbursement of thermal spa therapy by the national health insurance and the private health insurance companies of long COVID-19 patients that can consider, in a more refined way, the localization of patients and the thermal spa therapy offer. In this way, the cartographic study carried out here justifies the necessity to integrate geographical analysis in the modes of financing of the patients' care pathway in order to reduce the socio-economic inequalities of access to thermal spa therapy and to make this care a therapy accessible to the greatest number possible of patients.

## Contributions

The use of geographic mapping, allows the analysis of the integration of spa therapies in the care pathway of patients presenting symptoms of long COVID-19. The opening of and the integration of patient care pathways outside hospital establishments represents essential health stakes because they allow the reinforcement of care continuity [61] in a much more extended geographical perimeter. Taking the example of spa therapy, we present here the perspectives that geographic mapping offers to integrate spa therapies in the care pathway of long COVID-19 patients.

Firstly, on a theoretical level, this geographical mapping study shows that the potential attractiveness of spas areas and accessibility to thermal spas can determine the ways in which spa therapies can be integrated into the construction of territorialized, person-centered care pathways. The identification of these two geographical criteria represents a first conceptualization of the integration of a type of treatment such as spa therapy into a care pathway. This contribution deserves to be replicated for other types of treatment and other types of care pathways, but it already proposes a robust methodology to help researchers and practitioners measure the ways in which they can integrate territorial health services into a care pathway. Thus, in the case of spa therapies, "territorialized care pathways" [13] could emerge and constitute an opportunity to integrate the spa therapy offer in the downstream pathway of patients and, by the same token, contribute to a decompartmentalization of these two types of structures (hospitals and thermal spas). Indeed, the mobilization of Huff's method for evaluating the attractiveness of an area [31] makes it possible to define, on a practical level, in which French regions it is possible to implement these "territorialized care pathways" and in which regions the latter require the implementation of support mechanisms. Thus, using this spatial attractiveness method, we make this type of geographical analysis more dynamic by showing the extent to which a thermal spa is accessible to long COVID-19 patients.

Secondly, geographical mapping based on attractiveness of spa areas and accessibility of thermal spas is a managerial tool for reflection and dialogue between hospitals, doctors, spas, health and social protection authorities, with the aim of integrating a health service (in this case, spa therapy) into the patient's care pathway. In the case of the care of long COVID-19 patients, this mapping highlights the main criteria for reflection and discussion to integrate thermal care in territorialized care pathways for patients:

- continuity of the patient's care pathway and the perspectives offered by the patient's medical territory;

- possible clinical collaborations between hospitals and thermal spas located in the same geographical area;

- adapting of the methods of financing patients' care pathways according to their geographical location, mainly for patients who are farthest from thermal spas.

At a time when it has been demonstrated that thermal spas have a strong social and economic value for the region in which they are located [62], it is also important to show how these spas and their regions can be both accessible and attractive. The main criteria for integrating spa therapies into the care pathways of long COVID-19 patients, derived from our geographical mapping, are useful recommendations for improving this accessibility and attractiveness.

## Limitations

This mapping analysis of the management of long COVID-19 patients thus shows that the geographic approach makes it possible to inscribe patient care pathways in their territories for

care that is distant from the patient's location instead of limiting them to local clinical care. However, this is only a first analysis and, in future, it will be necessary to reinforce these results by mobilizing larger data concerning long COVID-19 patients treated in French thermal spas in order to verify the relevance of the managerial and health perspectives presented in this article.

## Conclusions and perspectives

From the example of the French thermal spa therapy offer, this article proposes to inscribe the care pathways of long COVID-19 patients in their territories instead of limiting them to clinical care in hospitals. Our research objective, which is to determine what place French thermal spa therapies can take in the care pathway of long COVID-19 patients, falls within the field of management and organization of care pathways for long COVID-19 patients, and in no way within the medical field. Through this study, geographic mapping analysis has proven to be a powerful analytical tool to reconcile the design of these pathways with the French thermal spa therapy offer. Firstly, the mapping analysis associated with the one of the attractiveness and accessibility to the thermal spas, which the Huff attractiveness method allows, makes it possible to describe the care pathways of the patients suffering from long COVID-19 in a dynamic way and not centered on a limited geographical area.

Secondly, these analyses then allow the identification of success and integration criteria of these care pathways in their territories adapted to the specificities of the produced care as well as to the geographical location of the care centers (here thermal) and of the potential patient base. This result then begins to demonstrate the original prospects offered by geographical attractiveness models (for example, Huff, [27, 56] to help managers manage and steer care pathways beyond the boundaries of their local areas. Of course, this type of spa treatment can only be implemented on a larger scale if the clinical benefits are further demonstrated by randomized trials.

## Supporting information

**S1 Appendix. Links to the public data used to produce Figs 1 to 3.**
(DOCX)

**S2 Appendix. Characteristics of the waters of spas specialized in ENT and respiratory, rheumatological and psychosomatic pathologies.**
(DOCX)

## Acknowledgments

Milhan Chaze, Laurent Mériade, Corinne Rochette, Mélina Bailly, Rea Bingula, Christelle Blavignac, Martine Duclos, Bertrand Evrard, Anne Cécile Fournier, Lena Pelissier and David Thivel form the complete membership of the "Immuno-Metabolic (IM) Phenotyping and management of COVID-19: Specificity of actors and of the Auvergne territory" (CAUVIM-19 research program). We would like to thank the InnovaTherm Cluster for its support.

## Author Contributions

**Conceptualization:** Milhan Chaze, Laurent Mériade, Corinne Rochette, Mélina Bailly, Rea Bingula, Christelle Blavignac, Martine Duclos, Bertrand Evrard, Anne Cécile Fournier, Lena Pelissier, David Thivel.

**Data curation:** David Thivel.

**Formal analysis:** Milhan Chaze.

**Funding acquisition:** Laurent Mériade, Corinne Rochette, Mélina Bailly, Rea Bingula, Christelle Blavignac, Martine Duclos, Bertrand Evrard, Anne Cécile Fournier, Lena Pelissier, David Thivel.

**Investigation:** Laurent Mériade, Corinne Rochette.

**Methodology:** Milhan Chaze, Laurent Mériade, Corinne Rochette.

**Project administration:** Laurent Mériade, Corinne Rochette, Mélina Bailly, Rea Bingula, Christelle Blavignac, Martine Duclos, Bertrand Evrard, Anne Cécile Fournier, Lena Pelissier, David Thivel.

**Software:** Milhan Chaze.

**Supervision:** Laurent Mériade, Corinne Rochette.

**Validation:** Milhan Chaze, Laurent Mériade, Corinne Rochette.

**Visualization:** Milhan Chaze, Laurent Mériade, Corinne Rochette.

**Writing – original draft:** Milhan Chaze, Corinne Rochette.

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
