## [Decision Letter · Decision Letter 0]

9 Nov 2023

PONE-D-23-18949Relying on the French territorial offer of thermal spa therapies to build a care pathway for long COVID-19 patientsPLOS ONE

Dear Dr. Mériade,

Thank you for submitting your manuscript to PLOS ONE. After careful consideration, we feel that it has merit but does not fully meet PLOS ONE’s publication criteria as it currently stands. Therefore, we invite you to submit a revised version of the manuscript that addresses the points raised during the review process.

ACADEMIC EDITOR

We greatly appreciate the effort, dedication and time you have put into the manuscript. After a thorough review by our panel of reviewers, there are several issues pointed out by them and a MAJOR REVISION is necessary before the article can be considered for publication.

The reviewers provided valuable feedback, and we ask that you carefully consider and address each point to enhance the quality and relevance of your work.

Additionally to reviewers comments, the following are mandatory revisions requested:

(i) the article lacks some clarity in the definition of the aims and the structure could be improved (e.g., there is some confusing between results and discussion section). Please ensure that your arguments and key points are presented logically and coherently throughout the article.

(ii) the article needs to deepen the literature review section, incorporating sufficient and relevant scholarly clinical studies associating thermalism and long COVID to provide a more robust background for your research.

(iii) the discussion and conclusion sections need more depth and precision. Enhance these sections by providing a more robust interpretation of the results and a clear conclusion that aligns with the objectives of your study.

Best regards,

We look forward to receiving your revised manuscript.

Kind regards,

Angela Mendes Freitas

Academic Editor

PLOS ONE

Journal Requirements:

"This publication is related to the project named “CAUVIM-19 – Immuno-Metabolic (IM) Phenotyping and management of COVID-19: Specificity of actors and of the Auvergne territory”. The “CAUVIM-19” project is co-funded by the FEDER (European Fund for Regional Development) as part of the European Union’s response to the COVID-19 pandemic"

6. We note that Figures 1, 2 and 3 in your submission contain map/satellite images which may be copyrighted. All PLOS content is published under the Creative Commons Attribution License (CC BY 4.0), which means that the manuscript, images, and Supporting Information files will be freely available online, and any third party is permitted to access, download, copy, distribute, and use these materials in any way, even commercially, with proper attribution. For these reasons, we cannot publish previously copyrighted maps or satellite images created using proprietary data, such as Google software (Google Maps, Street View, and Earth). For more information, see our copyright guidelines: http://journals.plos.org/plosone/s/licenses-and-copyright.

a. You may seek permission from the original copyright holder of Figures 1, 2 and 3 to publish the content specifically under the CC BY 4.0 license.  

Reviewers' comments:

Reviewer's Responses to Questions

**Comments to the Author**

1. Is the manuscript technically sound, and do the data support the conclusions?

Reviewer #1: Partly

Reviewer #2: Yes

Reviewer #3: No

2. Has the statistical analysis been performed appropriately and rigorously? 

Reviewer #1: Yes

Reviewer #2: Yes

Reviewer #3: No

3. Have the authors made all data underlying the findings in their manuscript fully available?

Reviewer #1: No

Reviewer #2: Yes

Reviewer #3: Yes

4. Is the manuscript presented in an intelligible fashion and written in standard English?

Reviewer #1: Yes

Reviewer #2: Yes

Reviewer #3: No

5. Review Comments to the Author

Reviewer #1: The authors address an important social issue: Long Covid. They propose to determine what place french thermal spa therapies can take in the care pathway of long COVID-19 patients through geographic mapping analysis.

I consider that although the text is very well written it is difficult to follow the research that the authors are presenting.

There are several structural aspects that could be used as examples, but the mais is related to the aim of the text: what are the authors addressing?. As example, the aim presented in the abstract is "determine what place french thermal spa therapies can take in the care pathway of long COVID-19 patients", latter it is read "evaluate the accessibility of each French department to thermal spas", in page 4 two questions are presented "How to integrate thermal spa therapy in the care pathway of long COVID-19 patients?" and "How to develop accessible care for patients by mobilizing national resources in thermal care?", on page 5 "The objective of this reading is, from the geographical cartography, to identify the perspectives offered by the territory to integrate thermal spa therapies in the care pathway of long COVID-l9 patients".

Considering this undefinition, and the originality of the text, it created some level of difficulty to follow the research line. I recommend the authors to make this more aligned and therefor clearer.. Before this rearrangement I cannot recommend this text for publication, neither offer a in-depth assessment

Some detailed aspects

Line 150 - reference is missing

Line 220 - Geographic mapping analysis - corresponds more to a Data subsection

Line 291 - usage of less scientific terminology. Further, the justification for the option is somehow weak. The best would be to do a stratified analysis

It is not clear if some spa did close in 2020 and did not open again, changing the offer

Generally, the results section merges results with discussion and with authors considerations about the topic

The resolution of the maps, at least i

---

## [Author Response · Author response to Decision Letter 0]

18 Jan 2024

Dear editors and reviewers, 

Thank you for your positive feedback. We have taken careful note of the reviewers' recommendations and have responded carefully to their comments by amending the text of the article (in blue) and responding individually to their requests for changes.

In the following lines, you will find your comments and the responses to these comments, mentioning the changes made to our article as well as the location of these changes. 

Thank you in advance for your attention to this new version of our article. 

Academic editor comments : 

We greatly appreciate the effort, dedication and time you have put into the manuscript. After a thorough review by our panel of reviewers, there are several issues pointed out by them and a MAJOR REVISION is necessary before the article can be considered for publication.

The reviewers provided valuable feedback, and we ask that you carefully consider and address each point to enhance the quality and relevance of your work.

Additionally to reviewers comments, the following are mandatory revisions requested:

(i) the article lacks some clarity in the definition of the aims and the structure could be improved (e.g., there is some confusing between results and discussion section). Please ensure that your arguments and key points are presented logically and coherently throughout the article.

(ii) the article needs to deepen the literature review section, incorporating sufficient and relevant scholarly clinical studies associating thermalism and long COVID to provide a more robust background for your research.

(iii) the discussion and conclusion sections need more depth and precision. Enhance these sections by providing a more robust interpretation of the results and a clear conclusion that aligns with the objectives of your study.

Answer : We would like to thank you for your editorial decision, which has given us the opportunity to perfect our work and submit an improved version of our manuscript. We therefore thank you for the rigour of your editorial process and the work of the reviewers, who have enabled us to respond to the mandatory revisions as follows: 

(i) as we mention in the responses to reviewers appearing below, we have added clarity to the research objective by mentioning throughout the text that the research objective is to "determine what place French thermal spa therapies can take in the care pathway of long COVID-19 patients". In addition, as suggested by reviewer 3, we have repositioned some results and methodological elements in the discussion section to ensure that our arguments and key points are presented in a logical and coherent manner throughout the article.

(ii) We have restructured the article by modifying the literature review section and incorporating sufficient scholarly and relevant clinical studies associating thermalism and long COVID. This now allows us to mention that there is no consensus concerning the benefits of thermal cures on the care of long COVID patients but that our research objective does not aim to settle this medical debate but to question the possibility of integrating these thermal cures into a care pathway by simply taking the example of the pathway of long COVID-19 patients. 

(iii) We have resized the discussion and conclusion sections by specifying once again that our results and contributions are above all managerial and organisational. This new structuring of the "discussion" and "conclusion" sections allows us to clearly show that our contributions are not medical but solely organisational and managerial and that they are aligned with the central objective of our study, which is "to determine what place French thermal spa therapies can take in the care pathway of long COVID-19 patients".

Reviewer #1: 

Remark 1 : The authors address an important social issue: Long Covid. They propose to determine what place french thermal spa therapies can take in the care pathway of long COVID-19 patients through geographic mapping analysis. I consider that although the text is very well written it is difficult to follow the research that the authors are presenting.

There are several structural aspects that could be used as examples, but the mais is related to the aim of the text: what are the authors addressing?. As example, the aim presented in the abstract is "determine what place french thermal spa therapies can take in the care pathway of long COVID-19 patients", latter it is read "evaluate the accessibility of each French department to thermal spas", in page 4 two questions are presented "How to integrate thermal spa therapy in the care pathway of long COVID-19 patients?" and "How to develop accessible care for patients by mobilizing national resources in thermal care?", on page 5 "The objective of this reading is, from the geographical cartography, to identify the perspectives offered by the territory to integrate thermal spa therapies in the care pathway of long COVID-l9 patients".

Considering this undefinition, and the originality of the text, it created some level of difficulty to follow the research line. I recommend the authors to make this more aligned and therefor clearer.. Before this rearrangement I cannot recommend this text for publication, neither offer a in-depth assessment

Answer 1 : Thank you for this comment, which allows us to see that we have undoubtedly introduced many questions into our text which make reading a little complicated. So, in the new manuscript, we have kept just one research question to "determine what place French thermal spa therapies can take in the care pathway of long COVID-19 patients" and we have therefore harmonised the text so that only this one question appears. On line 107 we now specify that the research question is "what place French thermal spa therapies can take in the care pathway of long COVID-19 patients". In addition, on lines 128-130, we state that "The objective of this reading is, from the geographical cartography, to determine what place French thermal spa therapies can take in the care pathway of long COVID-19 patients."

Remark 2 : Some detailed aspects

Line 150 - reference is missing

Line 220 - Geographic mapping analysis - corresponds more to a Data subsection

Line 291 - usage of less scientific terminology. Further, the justification for the option is somehow weak. The best would be to do a stratified analysis

Answer 2 : 

Line 158: We have added the reference in this new version [35]

Line 254: We have changed the title of the sub-part to improve the clarity of our remarks.

Line 334 and following: We opted for a group analysis of all spas, and not for a stratified analysis, in order to focus our discussion on the general accessibility of patients with COVID-19 to spas, without going into detail that would have led us to write an article much longer than the journal allows. However, we thank the reviewer for his suggestion: the application of a stratified analysis according to the types of pathologies managed in thermal spas could indeed make it possible to detail the results obtained in this article and be the subject of a subsequent publication. In order to explain our choice, we have added a justification sentence of the method.

Remark 3: It is not clear if some spa did close in 2020 and did not open again, changing the offer

Answer 3: after the state-ordered compulsory closure of spa resorts at the height of the COVID-19 outbreak, as confirmed by the “Conseil national des établissements thermaux” SPAs have all re-opened, sometimes a few weeks behind schedule. By 2023, some had not yet fully recovered their pre-COVID-19 attendance levels.

Remark 4 Generally, the results section merges results with discussion and with authors considerations about the topic

Answer 4 : In this new version, in order to clearly separate the results relating to the application of the model from their interpretation and consequences, we have rearranged the content of these two parts to distinguish them clearly. We have advanced the discussion section by starting it on line 432 and dividing it into four sub-sections (lines 433 et seq., lines 503 et seq., lines 533 et seq. and lines 557 et seq.).

Remark 5: The resolution of the maps, at least in the pdf form, is not good enough. The usage of black dots in a black to white map is not working

Answer 5 : Thank you for your comment. As in the previous version, we have supplied the maps in tiff image format, but it is possible that their transformation into pdf by the editorial manager does not provide good resolution. Nevertheless, by providing the maps in tiff image format as requested by the editorial manager, the resolution should be of good quality. In addition, in the maps (Figures 1 to 3), we have replaced the black dots with coloured dots. Thank you for this suggestion

Reviewer #2: 

The topic is very original and interesting. Studies as yours are very necessary, because it constitutes an important contribution to a gap in scientific literature. Furthermore, the authors have majestically combined disciplines such as geography or medicine, making it a quality multidisciplinary study. As suggestions:

Remark 1: -Check the use of the word "territory", it is not always translated that way into English. Maybe try others like "space"

Answer 1: Thank you for your comment. We have changed some of the words “territory” to a more appropriate synonym (area), particularly when it is associated with the notion of attractiveness.

Remark 2 -The Pinos & Shaw reference is wrong. The journal of publication was Tourism and Hospitality Research

Answer 2: Thank you very much for your review and your comments. We have changed the reference to include the exact name of the journal (line 732).

Reviewer #3: 

This study provides a geographical analysis of the relationship between the treatment needs of covid patients with persistent symptoms and the availability of spa treatments in France. It concludes with a call for spa treatments to be reimbursed by social security.

There are some major points that strongly penalize this article

Remark 1: The contribution and reimbursement of spa treatments are based on a demonstration of clinical benefit by means of a randomized comparative study. This has been done for the currently validated indications for spa treatments, such as osteoarthritis and venous insufficiency, to name but the most important, but there are no studies for covid and its after-effects. Even for recognized indications such as osteoarthritis, the debate on reimbursement remains wide open in France. For osteoarthritis, for example, we know that the benefit is modest and relates solely to quality of life (Ref 40 Forestier), i.e. a subjective criterion, while the studies are not blinded. For venous insufficiency (Ref Carpentier PH J Vasc Surg. 2014) with an objective clinical primary endpoint (incidence of post-cure venous ulcers), the study proved to be non-significant.

Answer 1: 

We thank you very much for this comment, which enables us to note that the opinions and studies which make it possible to account for the clinical benefit of thermal cures have difficulty in agreeing on the efficacy of these cures. In the new manuscript, we have tried to take account of the divided opinions concerning this question by integrating, for example, Ref Carpentier PH J Vasc Surg. 2014, which indicates modest benefits for venous insufficiency (lines 184-189). This allows us to mention how opinions and medical studies remain divided concerning the effectiveness of spa treatments (page 4). However, we also mention that we are not doctors and the aim of this research, as clearly stated in the abstract and introduction to the article, is to determine what place French thermal spa therapies can take in the care pathway of long COVID-19 patients. This means that we are studying, from an organisational point of view, how it would be possible to integrate spa treatments into the care pathway of long COVID patients, in parallel with more developed medical studies on the benefits of spa treatments. It turns out that in France, at the present time, many doctors prescribe these cures to long-standing COVID-19 patients and that the question of including these cures in the care pathway for long-standing COVID patients is a long one, even if there is no medical consensus on the actual benefits of these spa cures. We are social science researchers and we cannot comment on the medical effectiveness of these spa treatments, but we are trying to determine, from an organisational point of view, how they can be integrated into the care pathway for long COVID-19 patients. Also, in response to your very justified suggestions, we have tempered our argument by specifying on several occasions that even if medical studies suggest that there could be benefits from spa treatments for long COVID patients, there is no consensus or sufficient medical evidence (line 93-96 then lines 184-189). This allows us to specify that our objective is not to intervene in this medical debate but simply to determine in what ways it would be possible to integrate these cures into a pathway in order to respond more to an organisational and spatial issue than a medical one (lines 95-100).

Remark 2: The sentence "Important works have demonstrated, in different countries, that the medical management of COVID-19 patients favors the use of thermalism for its curative effects 92 related to post-disease pulmonary problems [8, 9] and for post-treatment rehabilitation [10, 11, 12]" is not properly supported and misleads the reader.

Answer 2 : We would like to thank you for this comment, which enabled us, in the new manuscript, to modify this argument by mentioning that even if certain studies reported a potential effect of spa treatments on the management of long COVID-19 patients, there was no medical consensus and that, while awaiting more in-depth and robust studies, we are content in this article to answer an organisational question aimed at determining what place spa treatments could take in the care pathway of long COVID-19 patients (lines 96-100). Also, as we indicate in the discussion of this article, our contributions are only organisational and managerial and in no way medical.

Remark 3: The studies cited in no way demonstrate the medical service rendered by a spa treatment for covid longus.

Reference 8 is a market analysis in Portugal with no clinical data.

Reference 9 merely suggests, by analogy with the respiratory indications of spa cures, that thermalism could, conditionally, be useful in covid without clinical data.

Reference 10 lists the possible actions of thermalism and suggests theoretically that it could be useful in covid.

Reference 11 is a letter to the editor suggesting theoretically that thermalism could have an effect on immunity.

Reference 12 is a playdate without any clinical data

Answer 3 :

Thank you for analysing these references using your medical expertise. As we have indicated, we are not seeking to demonstrate the benefits of spa treatments for the treatment of long-standing inflammatory bowel disease. We have found studies in the literature which suggest that spa treatments may have a positive effect on long-standing inflammatory bowel disease. We found fewer studies proving the contrary, but some studies show the limitations of spa therapies. This is why we have perhaps favoured studies mentioning the possible benefits of these cures and as we do not know the appropriate methods for carrying out these studies we have not been able to analyse their reliability precisely. We would therefore be grateful if you could share your analysis with us. Consequently, in the new manuscript, on your recommendations, we have presented the limits of this work, specifying that it could not yet allow us to postulate the efficacy of spa treatments in the care of long COVID-19. To this end, we have inserted new references presenting the limits of balneotherapy in medical treatment and the conditions in which it can be used (references 13, 14, 15, 16). We have also tried to evaluate the references presenting the benefits of s

---

## [Decision Letter · Decision Letter 1]

1 Mar 2024

PONE-D-23-18949R1Relying on the French territorial offer of thermal spa therapies to build a care pathway for long COVID-19 patientsPLOS ONE

Dear Dr. Mériade,

Thank you for submitting your manuscript to PLOS ONE. After careful consideration, we feel that it has merit but does not fully meet PLOS ONE’s publication criteria as it currently stands. Therefore, we invite you to submit a revised version of the manuscript that addresses the points raised during the review process.

**ACADEMIC EDITOR:**Dear Author,

In order to be published, there is need of making a minor revision as requested by one of the reviewers: the clear separation of discussion from results and inclusion of fig 3 reference in the result section.This is the last request of revision.Thank you!

We look forward to receiving your revised manuscript.

Kind regards,

Angela Mendes Freitas

Academic Editor

PLOS ONE

Journal Requirements:

Additional Editor Comments:

Dear Author,

In order to the article be published, there is need of making a minor revision as requested by one of the reviewers: the clear separation of discussion from results and incluion of fig 3 reference in the result section. This is a last revision request.

Thank you!

Reviewers' comments:

Reviewer's Responses to Questions

**Comments to the Author**

1. If the authors have adequately addressed your comments raised in a previous round of review and you feel that this manuscript is now acceptable for publication, you may indicate that here to bypass the “Comments to the Author” section, enter your conflict of interest statement in the “Confidential to Editor” section, and submit your "Accept" recommendation.

Reviewer #1: All comments have been addressed

Reviewer #2: All comments have been addressed

Reviewer #3: All comments have been addressed

2. Is the manuscript technically sound, and do the data support the conclusions?

Reviewer #1: Yes

Reviewer #2: (No Response)

Reviewer #3: No

3. Has the statistical analysis been performed appropriately and rigorously? 

Reviewer #1: Yes

Reviewer #2: (No Response)

Reviewer #3: I Don't Know

4. Have the authors made all data underlying the findings in their manuscript fully available?

Reviewer #1: Yes

Reviewer #2: (No Response)

Reviewer #3: Yes

5. Is the manuscript presented in an intelligible fashion and written in standard English?

Reviewer #1: Yes

Reviewer #2: (No Response)

Reviewer #3: Yes

6. Review Comments to the Author

Reviewer #1: The authors addressed the main issues and the text has quality to be published.

Nonetheless I still recommend to more clearly separate discussion from results and include fig 3 reference in the result section

Reviewer #2: (No Response)

Reviewer #3: The authors have partially modified their article, in particular on the key point: the absence of serious studies on the benefit-risk ratio of a spa treatment for long covid. Nevertheless, they continue to cast doubt by writing that there is no consensus on the efficacy of spa treatments. In France, this doubt persists for spa treatments evaluated by randomised trials (osteoarthritis, venous insufficiency, psoriasis, etc.) due to the methodological limitations of these studies. This is not true for long covid: as we have no studies other than marketing studies, or even studies currently being recruited, the consensus is clear: spa treatments have no place in the treatment of long covid (the HAS text, for example, never mentions spa treatments). This does not preclude a wellness stay at the expense of patients who so wish.

I will quote just two statements from the summary

1 "Work on long COVID-19 has mainly focused on clinical care in hospitals". This statement is false, as France's Haute Autorité de Santé (updated April 2023) recommends primary care management with specific advice for private practitioners and physiotherapists (https://www.has-sante.fr/jcms/p_3429665/en/symptomes-prolonges-de-la-covid-19-dit-covid-long-la-has-actualise-ses-travaux).

2 "Thermal spa therapies represent a therapeutic offer outside of health care institutions that are nationally or even internationally attractive". In what way are spa treatments provided by specialist doctors (osteoarthritis, venous insufficiency or dermatology for the majority) with the help of physiotherapists and nurses outside health care institutions? Spa treatments are reimbursed by the French social security system, pathology by pathology, when the spas have provided scientific proof, through prospective studies, of the favourable benefit/risk ratio of a spa cure. This is not the case for long covid, where no study has provided proof of efficacy or even feasibility. We can legitimately question the tolerance of the cure by long covid patients (risk of post-exertional symptoms, for example).

The authors argue that they are not doctors but sociologists and present a method for organising care. This method is potentially interesting but why apply it to this case of covid long and thermalism ?

Why talk about all the spas without distinguishing their specific indications ? What doctor would refer these patients to a spa specialising in dermatology, with specific dermatology treatments, to treat a long covid with respiratory difficulties ?

In conclusion, the authors should submit their work to scientific journals in their field of sociology or choose a recognised indication of thermalism.

7. PLOS authors have the option to publish the peer review history of their article (what does this mean?). If published, this will include your full peer review and any attached files.

Reviewer #1: No

Reviewer #2: No

Reviewer #3: **Yes: **BOSSON Jean Luc

---

## [Author Response · Author response to Decision Letter 1]

19 Mar 2024

Dear editor,

Thank you for your positive feedback. We have taken careful note of your recommendations and have responded to your requests for changes by modifying the text of the article (in blue in the revised manuscript with track changes).

In the following lines, you will find the answers to your comments, mentioning the changes made to our article as well as the location of these changes. 

Academic editor feedback : 

"In order to be published, there is need of making a minor revision as requested by one of the reviewers: the clear separation of discussion from results and inclusion of fig 3 reference in the result section. This is the last request of revision."

Answer: 

Thank you very much for your review requests, which have enabled us to further improve the structure of our article. To this end, as requested by one of the reviewers, we have included fig. 3 in the result section (line 440). For greater coherence, we have also inserted the section relating to this fig. 3 in the result section (lines 432 to 471). 

To provide a clearer separation between the result section and the discussion section, we've inserted an introductory paragraph to the discussion section (lines 472 to 475) as well as a new title and introductory sentence to introduce this section (lines 479 to 482).

Thank you again for your comments and suggestions, and for taking the time to read this new version of our article. 

Yours faithfully

The authors

---

## [Editor Report · Decision Letter 2]

2 Apr 2024

Relying on the French territorial offer of thermal spa therapies to build a care pathway for long COVID-19 patients

PONE-D-23-18949R2

Dear Dr. Mériade,

We’re pleased to inform you that your manuscript has been judged scientifically suitable for publication and will be formally accepted for publication once it meets all outstanding technical requirements.

Kind regards,

Angela Mendes Freitas

Academic Editor

PLOS ONE
---

## [Editor Report · Acceptance letter]

8 Apr 2024

PONE-D-23-18949R2 

PLOS ONE

Dear Dr. Mériade, 

I'm pleased to inform you that your manuscript has been deemed suitable for publication in PLOS ONE. Congratulations! Your manuscript is now being handed over to our production team.

Kind regards, 

on behalf of

Dr. Angela Mendes Freitas 

Academic Editor

PLOS ONE